# Validation of the COVID-19 Digital Health Literacy Instrument in the Italian Language: A Cross-Sectional Study of Italian University Students

**DOI:** 10.3390/ijerph19106247

**Published:** 2022-05-20

**Authors:** Chiara Lorini, Veronica Velasco, Guglielmo Bonaccorsi, Kevin Dadaczynski, Orkan Okan, Patrizio Zanobini, Luca P. Vecchio

**Affiliations:** 1Department of Health Science, University of Florence, 50134 Florence, Italy; chiara.lorini@unifi.it (C.L.); guglielmo.bonaccorsi@unifi.it (G.B.); patrizio.zanobini@unifi.it (P.Z.); 2Department of Psychology, University of Milano-Bicocca, 20123 Milan, Italy; luca.vecchio@unimib.it; 3Department of Health Science, Fulda University of Applied Sciences, 36037 Fulda, Germany; kevin.dadaczynski@pg.hs-fulda.de; 4Centre for Applied Health Science, Leuphana University Lueneburg, 21335 Lueneburg, Germany; 5Department of Sports and Health Science, Technical University Munich, 80992 Munich, Germany; orkan.okan@tum.de

**Keywords:** digital health literacy, COVID-19, university students, infodemic, measurement, scale validation

## Abstract

The Coronavirus Disease 19 (COVID-19) pandemic and the associated “infodemic” have shown the importance of surveillance and promotion of health literacy, especially for young adults such as university students who use digital media to a very high degree. This study aimed to assess the validity and reliability of the Italian version of the COVID-19 adapted version of the Digital Health Literacy Instrument (DHLI). This cross-sectional study is part of the COVID-19 University Students Survey involving 3985 students from two Italian universities. First, item analysis and internal consistency were assessed. Then, Principal Component Analysis (PCA) and Confirmatory Factor Analyses (CFA) were performed comparing different models. The Italian DHLI showed good psychometric characteristics. The protecting privacy subscale was excluded, given the criticalities presented in the validation process. CFA confirmed the four-factor structure, also including a high-order factor. This result allows using the scale to measure a global level of digital health literacy and consider its levels separately for each construct component: searching the web for information, evaluating reliability, determining personal relevance, and adding self-generated content.

## 1. Introduction

### 1.1. Health Literacy and COVID-19

The Coronavirus Disease 19 (COVID-19) pandemic has negatively impacted many areas of people’s lives, health, and wellbeing. Although the pandemic situation appears to be improving in many countries, the situation is still critical, and many health-related decisions must be made by individuals, organizations, and governments. Many authors and international institutions argue that the pandemic is accompanied and exacerbated by an “infodemic”, a global epidemic of information, both accurate information and misinformation that spreads rapidly through social media and other platforms [1,2,3,4,5,6,7]. This misinformation has affected individual protective behaviors and endangered the government and health authorities’ efforts to manage COVID-19 [7,8]. Moreover, the massive amount of information about COVID-19 can increase citizens’ anxiety and uncertainty [7]. In the long term, it may reduce the credibility of scientific expertise, with consequences in many health areas [9]. The importance of the media in disseminating health information and influencing health-related decisions is not a new phenomenon, and it has been increasing in the last 20 years [10,11]. In 2014, the Eurobarometer results showed that six out of ten Europeans went online when looking for health information [12]. The Eurostat Information and Communication Technologies (ICT) household survey showed that in the first quarter of 2020 in Europe, the percentage reached 80% for some countries [11]. During the pandemic, the HLS-COVID-19 survey reported similar results in several countries [3].

Some strategies to fight an infodemic have been proposed. Eysenbach [4] suggested four pillars: (1) information monitoring (infoveillance); (2) building eHealth Literacy and science literacy capacity; (3) encouraging knowledge refinement and quality improvement processes such as fact-checking and peer-review; and (4) accurate and timely knowledge translation, minimizing distorting factors such as political or commercial influences. A World Health Organization (WHO) technical consultation identified six policy implications [13]: (1) base interventions and messages on science and evidence; (2) translate knowledge into understandable and actionable behavior-change messages; (3) collect community needs and tailor messages; (4) establish intersectoral partnerships to amplify information impact; (5) inform health authorities actions with reliable information and adapt them to the circulating narratives; (6) further develop infodemic management through data science, socio-behavioral, and other research. Many of these strategies are related to a crucial skill necessary to face an infodemic: health literacy (HL; see Appendix A for acronyms’ definition). This construct has been defined by the [14] as “the cognitive and social skills which determine the motivation and ability of individuals to gain access to, understand and use information in ways which promote and maintain good health”. HL is considered an important determinant of health by many international bodies. It has been shown to influence healthy behaviors, health and social services access, health outcomes, health-related inequalities, the ability to manage long-term healthcare conditions, and social capital [15,16,17,18,19,20,21]. Low HL is also associated with reduced vaccination adherence [22,23]. During the current emergency related to COVID-19, HL can increase the likelihood of being well-informed and aware of risks, recognizing resources, and following recommendations. It may promote the adoption of protective and preventive behaviors and contribute to inequity reduction and prevention. This capacity is also needed to manage the vast amount of information, changing and conflicting about health and the epidemiological situation. It could help citizens to understand the reasons behind the norms and recommendations and to foster social responsibility [7,24,25,26,27,28]. Efforts are needed to assess HL and digital health literacy (DHL), identify population needs, and suggest inputs for policy and intervention developments to manage the COVID-19 pandemic [7,26,29].

### 1.2. Digital Health Literacy Assessment

Given the critical role of the media in disseminating health information, it is particularly important to consider eHL, DHL, and media HL. These terms have been used in the literature considering HL use with information from media, electronic sources, and communication technologies [30,31,32]. The studies about DHL showed that self-perceived skills to use online information influence people’s health, health care quality, health outcomes, and vaccination adherence [33,34,35]. However, the measurement of DHL is still critical. This construct includes several components, and the more complex levels of HL are often neglected in most of the assessment tools [30]. Moreover, media and technologies change continuously, and updated measures of DHL should consider a broad spectrum of applications [30,36,37,38,39]. The most used HL measurement in this area is the eHealth Literacy Scale (eHEALS) by Norman and Skinner [32]. This scale presents several criticalities [36,40,41,42,43]. First, its validity is not clear. Second, it focuses on seeking and appraising online information, but it does not address critical and interactive health literacy. Third, it does not consider the new tools provided by the internet and technologies. A more complex scale recently developed to assess digital health literacy is the Digital Health Literacy Instrument (DHLI) [36]. It aims to incorporate the skills necessary to use the broad spectrum of applications offered by the internet and communication technologies and give valid information about people’s actual competence level. However, this scale has only been used with a Dutch study population. Moreover, the COVID-19 pandemic brought new challenges in the digital health literacy measurements requiring specific instruments about COVID-19 information. New validated and cross-national measurements are needed.

A particularly important target during the COVID-19 pandemic consists of university students. The emergency has affected the university organization with a high impact on students [44,45]. University students represent a huge proportion of young people in most European countries [46]. They belong to a constantly connected generation with frequent access to technologies [47]. Many studies showed high use of the internet to search for health information, referring to both reliable and not reliable sources [6,44,48,49,50,51,52]. Moreover, university students can be considered a vulnerable population to be monitored. Entering the university represents a transition period characterized by new experiences and challenging circumstances to cope with. In some cases, it also coincides with leaving the parental home. Students must integrate themselves into a new social environment characterized by higher academic demands, lower levels of structure, and greater autonomy [53]. Previous studies about university students’ HL showed that this population has low levels of eHL skills [54]. Moreover, some studies underlined the importance of considering inequalities among this population, showing the relevant effect of demographic characteristics and social-economic levels on health literacy and students’ wellbeing [50,55,56,57]. At the same time, university students may represent a crucial social resource. In fact, they are a young segment of the population, have low levels of chronic diseases, and can access many academic inputs and support. For these reasons, they can also be considered a secondary target aimed at increasing health literacy levels among the general population and activating literate and community processes.

The assessment of DHL among university students during the pandemic is particularly important in Italy. Italy was one of the countries most affected by the COVID-19 pandemic, mainly during the first wave. National and regional governments had to face the pandemic when there was little available information, and few scientific studies were realized. This condition may increase the dissemination of conflicting information and the spread of the infodemic. Moreover, previous cross-national studies about HL showed that Italian adolescents and adults reported lower levels of HL compared with other European countries [58,59,60,61]. These results show the importance of health literacy monitoring to provide inputs for policy and intervention design and developing valid and updated Italian tools to assess health literacy and digital health literacy. Finally, more efforts are needed to promote university students’ health and wellbeing in Italy. The Eurostat study [62] showed that Italy is the country with the second-highest levels of university drop-out in Europe, showing the vulnerability of this population.

The aim of this study is to validate the Italian version of the DHLI developed by van der Vaart and Drossaert [36] and adapted to the COVID-19 pandemic by Dadaczynski and colleagues [63]. This scale has been included in the University survey of the COVID-HL network, and it showed good reliability and validity properties in other countries [6,50]. To our knowledge, only the Portuguese version of the scale has been validated [64].

## 2. Materials and Methods

### 2.1. Method for Survey

This cross-sectional study is part of the COVID-19 University Students Survey, conducted in 28 Countries by the COVID-HL Network [63,65].

The study was designed and conducted in accordance with the Helsinki declaration and was approved by both the Ethics Commission of the University of Florence (n. 108, 07/07/2020) and by the Ethics Committee of the Psychology Department of the Milano-Bicocca University (n. RM-2020-284, 17/04/2020).

Two Italian Universities—Milan-Bicocca and Florence—participated in the survey and collected data, respectively, from 6 May to 9 June 2020 and from 17 August to 3 October 2020. The first period corresponded to the end of the first wave of the COVID-19 pandemic in Italy, when the spread of infection was particularly high in Lombardy (where Milan is located), and a national lockdown was put in place (Italian universities stopped all in-person activities) [66,67]. The second period corresponded to the beginning of the second wave—which strongly affected all Italian regions—with the introduction of many restrictive measures (also at the universities) according to the degree of diffusion of the infection. In Italy, as of 3 October 2020, 322,983 COVID-19 cases and 35,986 COVID-19 deaths were registered [68].

### 2.2. Data Collection

A convenience sample of Italian university students was used, involving two Italian universities: Milan-Bicocca and Florence. The online questionnaire was shared with university students from all the courses (bachelor, master, Ph.D.) by institutional email from the university, university social media, and through the involvement of the students’ representatives. A reminder was sent one week after the survey started and after a further two weeks. The survey was distributed using the platforms Qualtrics (for Milano-Bicocca University) and Google (for Florence University). Participation in the study was voluntary, and anonymity was ensured.

University students enrolled at the university were eligible to participate in this study. Respondents were initially asked to indicate their current status. Those who indicated that they were not currently enrolled as students were excluded from the data set.

In 2020, there were about 33,000 students enrolled at the University of Milan-Bicocca and about 50,000 at the University of Florence. In total, a convenience sample of 3985 students fulfilled the online questionnaire after having completed informed written consent.

### 2.3. Development of the Survey Form

Data were collected using a questionnaire developed by Dadaczynski and colleagues [63] and included existing validated scales adapted to the COVID-19 pandemic and newly developed scales. In particular, the following data were collected: sociodemographic information; life situation and future anxiety; DHL and information-seeking behavior; personal health situation.

The translation of the COVID-19 DHLI into the Italian language followed the standard procedure [69]. Two independent native English speakers translated the questionnaire into the Italian language, and then two independent native Italian speakers back-translated the two versions into English. The four versions (two in Italian and two in English) were assessed and discussed by the research group to identify any discrepancies emerging from the process and to reach a shared final version of the COVID-19 DHLI in the Italian language.

In Appendix A, the final version of the COVID-19 DHLI in Italian language is reported, as well as the English version.

### 2.4. COVID-19 Digital Health Literacy Instrument

DHL related to the COVID-19 pandemic was measured by adapting the DHLI developed by van der Vaart and Drossaert [36]. The original DHLI was composed of seven subscales (operational skills, navigation skills, information searching, evaluating reliability, determining relevance, adding self-generated context, protecting privacy), each including three items to be answered on a 4-point Likert scale. In the validation study, subscores were calculated as the mean score by subscales, and a total score was calculated by using the total mean, for which answers on at least 18 items (85.7% of the total items) were necessary. The original DHLI showed good psychometric characteristics, either when considering the total scale, or considering each subscale, except for the “protecting privacy” subscale.

To generate the COVID-19 DHLI, five out of seven subscales were included and modified with respect to COVID-19: searching the web for information on COVID-19 (DHLIsearch); adding self-generated content on COVID-19 (DHLIcont); evaluating the reliability of COVID-19-related information (DHLIrely); determining personal relevance of COVID-19-related information (DHLIrelev); protecting privacy on the internet (DHLIpriv) [44]. For the first four subscales, response options were “very difficult” (1), “difficult” (2), “easy” (3), and “very easy” (4), while for the DHLIpriv subscale, they were “often” (1), “several times” (2), “once” (3), and “never” (4) (Appendix A).

### 2.5. Statistical Analysis

Data were presented as a percentage, mean (±standard deviation—SD), and median (interquartile range—IQR). The item, subscale, and scale scores were tested for normality using the Kolmogorov–Smirnov test.

For the scale evaluation in the Italian language, the classical measurement theory was applied to assess the reliability and the validity [70,71]. In assessing the psychometric properties of the instrument, we assumed that DHL—as HL, from which it derives—is a multidimensional construct. Based on this assumption, the following four steps were considered.

First, we performed item analysis to examine: (i) the distribution of the responses, to determine the percentage of missing items (that is, a proxy of item difficulties and comprehensibility) and the presence of a ceiling or floor effect (that is, a limit in variability due to an excess—at least 20%- of responses for the highest or the lowest category, respectively); (ii) the correlation between the items, measured using Pearson or Spearman correlation analysis, as appropriate.

Second, the internal consistency (i.e., the degree to which the respondents answered consistently) was assessed by calculating Cronbach’s alpha (a measure of reliability). In particular, Cronbach’s alpha was calculated for the entire scale, for the entire scale excluding the DHLI privacy items [44], for the entire scale if each item was deleted, and by subscales. Acceptable values of the alpha range from 0.70 to 0.90, indicating the items explain the same underlying concept or construct without redundancies [72,73].

Third, a Principal Component Analysis (PCA) with varimax rotation was applied to determine the number of factors that fit the data. In this perspective, different approaches and models were applied: (i) two PCA models to assess the components according to the eigenvalues (i.e., fitting the data), including (Model 1) or excluding (Model 2) the DHLI privacy items, respectively; (ii) two PCA models to assess the components according to the number of expected components, including (Model 3, with five components) or excluding (Model 4, with four components) the DHLI privacy items, respectively. For each model, the explained variance, the Kaiser–Meyer–Olkin test (KMO, should exceed 0.80 for the PCA results and the multidimensional components to be reliable), and the Barlett sphericity test were used to determine the goodness of the models [74,75].

Fourth, Confirmatory Factor Analyses (CFA) were performed to confirm the dimensions of the scale. Based on the results of the PCA, the DHLI privacy items were excluded from the analysis. Three models were compared: (i) a model was applied to test unidimensionality (Model A); (ii) one model was performed to identify the subscales (Models B); (iii) one model was added to test for a high-order factor (Model C). The CFA results were evaluated by using several fit indices related to the overall model fit, model comparison, and model parsimony [76,77,78,79]. As regards to the overall model fit, Root Mean Square Error of Approximation (RMSEA) values lower than 0.05 are usually considered good, while values lower than 0.08 are considered acceptable, Standardized Root Mean Square Residual (SRMR) values lower than 0.08 are usually considered good, while values lower than 0.10 are considered acceptable. The following model comparisons were used: the Goodness-of-Fit Index (GFI), the Comparative Fit Index (CFI), and the Non-normed Fit Index (NNFI). For all indexes, values equal to or higher than 0.90 are considered acceptable, while values equal to or higher than 0.95 are considered good. Finally, model parsimony was assessed as a criterion for choosing between alternative models. The Parsimony Normed Fit Index (PNFI) and the Parsimony Goodness-of-Fit Index (PGFI) were used. Higher PNFI and PGFI indicate a more parsimonious fit.

Then, the subscales and scale scores were calculated according to the results of the PCA and of the CFA. For each subscale, as well as for the entire scale, the scores were calculated as the mean value of the scores reported for each included item. According to what was described for other instruments developed to measure HL, the scores were calculated if the number of missing answers was not too high [80,81,82]. In particular, a cut-off value of 1 missing response for the subscale (67% of filled-in items) and of 4 responses for the entire scale (excluding DHLI privacy items) (67% of filled-in items) was adopted. To complete the scale evaluation, a correlation analysis—Pearson or Spearman, as appropriate—was performed between the subscale and scale scores.

For each analysis, an alfa level of 0.05 was considered significant. The analyses were conducted using IBM SPSS Statistics 27 and Lisrel 8.

## 3. Results

### 3.1. Description of the Sample

Since 960 students declared to have not searched the internet in the last four weeks for information about COVID-19 and so can incur a recall bias, the analyses for the validation of the COVID-19 DHLI in the Italian language were conducted on a subsample of 3025 students, namely those who declared having searched the internet in the last four weeks for information about COVID-19, either for themselves or for other people. They were mostly females (71.1%), attending a bachelor’s program (60.6%), defining sufficient or completely sufficient their financial situation (70.7%), and economically supported by parents (83.8%). Their mean age was 23.1 years (±5.0 years), and their median age was 22 years (IQR: 20–24 years). The mean number of semesters at the university was 5.7 (±4.8), while the median value was 5 (IQR: 2–8).

### 3.2. COVID-19 DHLI in the Italian Language: Items Responses and Correlation Analysis

Table 1 reports the item responses. For each questionnaire, the number of missing responses varied from 0 (*n* = 2668; 88.2%) to 15 (*n* = 29; 1%). For 94.2%, the number of missing responses was equal to or less than 5 (i.e., equal to or less than 33.3% of the entire DHLI). The higher percentage of missing values was for the items of the protecting privacy subscale (DHLIpriv).

A ceiling effect (i.e., more of the 20% of the responses for the “very easy” or “never” options) was observed for one item of the information searching subscale (DHLIsearch2—*When you search the Internet for information on coronavirus or related topics, how easy or difficult is it for you to use the proper words or search query to find the information you are looking for?)*, for one of the evaluation reliability subscale (DHLIrely3—*When you search the Internet for information on the coronavirus or related topics, how easy or difficult is it for you to check different websites to see whether they provide the same information?),* and for the three items of the protecting privacy subscales (DHLIpriv1, DHLIpriv2, DHLIpriv3).

For all the items, the responses were not normally distributed. All the items were significantly correlated with each other, except for DHLIpriv 1 with DHLIpriv2 and DHLIpriv3 (Table 2). When statistically significant associations were observed, rho values varied from 0.04 to 0.68. The highest rho values (rho ≥ 0.4) were observed: (i) between the items of the same subscale; (ii) between DHLIrelev1 and all the items of both the searching and relevance DHLI subscales; and (iii) between DHLIrely1 and DHLIsearch1 and DHLIsearch3. For the items of the DHLIpriv subscale, low rho values were observed, except for the correlation between DHLIpriv2 and DHLIpriv3.

### 3.3. COVID-19 DHLI in the Italian Language: Reliability and Principal Component Analysis (PCA)

Cronbach’s alpha revealed a good internal consistency, either for the entire scale (also if a single item was deleted, as well as including or excluding DHLIpriv) or by subscales (alpha values between 0.7 and 0.9), except for the DHLI privacy subscale (alpha = 0.392) (Table 3). In particular, Cronbach’s alpha was higher when DHLI privacy items (the entire scale or each item) were deleted.

At the PCA analysis, all the tested models presented good fit indexes (Table 4), with better values for Model 4 (with four components—as expected by literature excluding DHLIpriv items). When considering the eigenvalues (i.e., fitting the data, Model 1 and 2) as well as when assessing the components according to the number of expected components, including DHLIpriv (Model 3), two items of the DHLIrely subscale (DHLIrely2 and DHLIrely3) and the three items of the DHLIrelev subscale converged in the same component, while DHLIrely1 was included in the same component of the DHLIsearch subscale. Differently, when assessing the components according to the number of expected components excluding DHLIpriv (i.e., four), the items converged in the four subscales as expected.

### 3.4. COVID-19 DHLI in the Italian Language: Confirmatory Factor Analysis (CFA)

The results indicated that Model B and C fitted the data well and better than the unidimensional Model A. All indexes’ values of both Model B and C are good, confirming the four subscales reported in the literature and emerged by the PCA. Comparing Models B and C, the parsimony indexes showed better values for Model C. This model includes a high-order factor (Table 5, Figure 1).

### 3.5. COVID-19 DHLI Subscales and Scale Scores

For each subscale, the scores were calculated as the mean of the single items, excluding the subjects with more than one missing value. The score for the entire scale (excluding DHLI privacy items) was calculated as the mean of the single items, excluding those with more than four missing values.

The score distribution was very similar for the subscales, as well as for the DHLI scale score (excluding DHLIpriv) (Figure 2). The subscales and scale scores were significantly correlated (*p* < 0.01), with rho values higher than 0.40 (range: 0.40–0.55) between subscales and higher than 0.70 (range: 0.73–0.80) between each subscale and the total score.

## 4. Discussion

The COVID-19 pandemic had (and still has) a tremendous impact on many aspects of personal and social life in Europe and worldwide. Italy (especially northern Italy) was the first in Europe to face the massive spread of SARS-CoV-2. At the onset of the pandemic, knowledge about the consequences was unclear, and strategies to prevent the infection were still ill-defined. The uncertainty of the situation allowed misinformation to develop and circulate rapidly, assuming the characteristics of an “infodemic”, further amplified by people’s easy access to media and the use of the internet as the primary source of health-related facts and data. Conflicting or wrong information about COVID-19 favored the affirmation of beliefs about ineffective protective and preventive behaviors or inappropriate therapeutic methods, up to the point of refusing to acknowledge the existence of the health emergency (see, for example, the diffusion of the so-called “no vax movement”).

Promoting health literacy (HL) could be an essential strategy to combat the infodemic. It can increase the likelihood of being well-informed and aware of risks, identifying resources, and critically evaluating the changing and conflicting information about the epidemiological situation [3]. To this end, it is crucial to have valid and reliable tools to assess HL and digital HL (DHL) in particular. Among the measures of DHL, we took into consideration the DHLI developed by van der Vaart and Drossaert [36] and adapted to the COVID-19 pandemic by Dadaczynski and colleagues [63]. It is a theory-based and comprehensive instrument to measure personal health literacy, primarily referring to online health information and healthcare-related digital applications. This scale was used in the COVID-19 University Student Survey conducted by the COVID-HL research Consortium, allowing comparisons of the DHL levels among the university student population, recognized as an interesting target for assessing and monitoring DHL.

This study aimed to assess the validity and reliability of the Italian version of the DHLI scale. Based on our results, the Italian DHLI shows good psychometric characteristics.

The number of missing responses was very low, meaning no problems comprehending the scale. The great majority of the valid responses recorded in our study (*n* = 3025) had no missing values for the DHLI scale (*n* = 2668; 88.2%). For each item of the scale, the percentage of missings remained rather low, representing in most cases less than 3% of the responses. Only for the self-generated contents subscale (DHLIcont) this percentage slightly increased (around 6%). At the same time, it reached the highest values for the three items of the protecting privacy subscale (DHLIpriv), where the percentages of missing responses are between 8.4% and 9.0%.

For four out of the five subscales of the DHLI, the distribution of the responses covers all the response options adequately, with no floor or ceiling effects, showing that the instrument is good enough to assess the variability of the phenomenon. However, for all the items of the privacy subscale, we observed an excess of responses (more than 40%) for the highest category (“never”), which in one case—concerning the sharing of someone else’s private information—amount to 80%. This result, along with the previous one concerning missing data, reveals some criticalities for this subscale.

Construct validity, as revealed by correlation analyses, appears adequate. The highest correlation values were observed between the items of the same subscales. It is interesting to note that one item of the relevance subscale (DHLIrelev1—When you search the internet for information on the coronavirus or related topics, how easy or difficult is it for you to decide if the information you found is applicable to you?) is strongly associated with all the items of both the information searching subscale and the evaluation reliability subscale. Indeed, the wording of the item refers to decision-making or evaluative processes that also characterize the associated items. Once again, items of the privacy subscale behave differently. Indeed, inter-item correlations are very low in this case, as for all the other items of the entire scale. The associations with some items of the other scales are not statistically significant for two items.

Given the weakness of the protecting privacy subscale, we decided not to consider it in the Italian version of DHLI. The reliability analysis supports our decision. Indeed, Cronbach’s alfa for the full-scale increases when excluding the subscale items. Regarding the reliability of each subscale, alfa values are acceptable for all of them (ranging from 0.74 to 0.83) except for the privacy one (0.39). Problems with the privacy subscale resemble those already reported by other authors. Indeed, even in the original paper on the DHLI development, the reliability of the privacy subscale was unsatisfactory [36]. The authors recognize that the subscale should be further improved. The same conclusion is proposed by Dadaczynski et al. [44] in their study about DHL among university students during Germany’s first wave of COVID-19. Moreover, a recent study aimed at validating the DHLI for Portuguese university students also excluded the privacy subscale due to psychometric problems [64].

As far as it concerns the structural components of the DHLI, the results obtained in the factor analyses once again sustain the exclusion of the protecting privacy subscale. Among the four PCA models we applied to determine the number of factors that fit the data, we obtained better values for Model 4, the model run according to the number of expected components. The variance explained was the highest, and the results were coherent with the theoretical assumptions of the construct. However, it is noteworthy to consider the alternative structure obtained from PCA based on eigenvalues while still excluding the privacy items. In this case, a three-factor solution emerged. The three items of the information searching subscale constitute one factor. A second factor includes the items of the adding self-generated content subscale. While in the third factor, the items of the reliability and relevance subscales collapse. Indeed, this result seems to differentiate between a “pragmatic” component of DHL—concerning the ability to find the correct information and correctly express one’s point of view—and an “evaluative” component about the adequacy of the retrieved information. Although it may be interesting to consider this distinction, the four-factor model still represents the better solution. Indeed, apart from presenting a cleaner structure than other solutions, it ensures comparability with cross-cultural studies conducted in other countries.

CFA confirms the four-factor structure, also including a high-order factor. Indeed, the 4 + 1 model (Model C) showed better parsimony indexes (PNFI = 0.7379; PGFI = 0.6121) compared to the four-factor model (Model B) with no higher-order factor (PNFI = 0.7093; PGFI = 0.5891). This result allows using the scale to measure a global level of DHL and consider DHL levels separately for each of the construct components.

The validation of the scale allowed for identifying digital health literacy levels among the university students participating in the study. Overall, these levels are relatively low, hovering around the central theoretical value of the scale (equal to 2.5). Mean scores are always less than 3, corresponding to the “easy” option of the answer categories, both in every subscale and for the entire scale. This result should get attention, given the young age of the participants and their presumably high-frequency use of the internet to search for information. However, other studies concerning health literacy in Italy reported low levels, both considering the general population [83] and younger students [58].

However, it needs to be emphasized that our participants do not constitute a representative sample of the Italian university student population. This represents the main limitation of our study since the results cannot be generalized. Moreover, given the convenience sampling procedure used, we must be aware of selection bias. At the same time, we must recognize that many of the university students who participated in the study belong to two important universities in northern and central Italy. Both are “general” universities, offering study courses belonging to all the major disciplines in which students from all over Italy enroll. Indeed, according to Dadaczynski et al.’s [44] results, sociodemographic characteristics do not seem to have an important influence on DHL levels (except for gender). So, the numerosity of our sample can be considered a reasonable guarantee that our results are adequate. In any case, conducting a survey about DHL on a representative sample of the general population can be a further step to be considered.

A limitation of the study concerns the use of self-assessment to measure DHL. In fact, what people think they know does not always correspond to what they actually know. People tend to be overconfident or underconfident as a consequence of the matching between knowledge, confidence, self-efficacy, and emotional distress. Overconfidence or underconfidence may differ from country to country, as they are also influenced by cultural factors [61], and this may have affected the validity of the DHLI. In fact, a previous study conducted in Florence in a sample of the general population has described that the level of health literacy measured using a self-assessed tool tends to be higher than those measured using an objective tool and that discrepancy tends to be higher among younger people [61]. Future studies on the effect of overestimation or underestimation on the validity of self-assessed measures of DHL (and more generally of HL) could shed light and are to be encouraged.

## 5. Conclusions

This study presents the Italian validation of the COVID-19 DHLI, adapted by Dadaczynski and colleagues [63] from the DHLI developed by van der Vaart and Drossaert [36]. The scale comprehends four subscales referring to four important skills for dealing with web-based COVID-19-related health data: (1) searching the web for information; (2) evaluating their reliability; (3) personal relevance; and (4) adding self-generated content. We excluded the protecting privacy sub-scale, given the criticalities presented in the validation process concerning its reliability and our participants’ adequate interpretation.

The scale can be used both to assess the general competence of DHL and to evaluate the levels of DHL in specific skills. This can be very useful in informing the development of health promotion programs to improve health literacy capacities. At the same time, having a global measure of DHL is important for developing health communication tailored to the assessed level of the target public you want to address. Both pieces of information are crucial for the strategies to contrast the COVID-19 pandemic in order to improve their efficacy and reduce the influence of the “infodemic” in promoting inadequate behaviors to face health threats.

## Figures and Tables

**Figure 1 ijerph-19-06247-f001:**
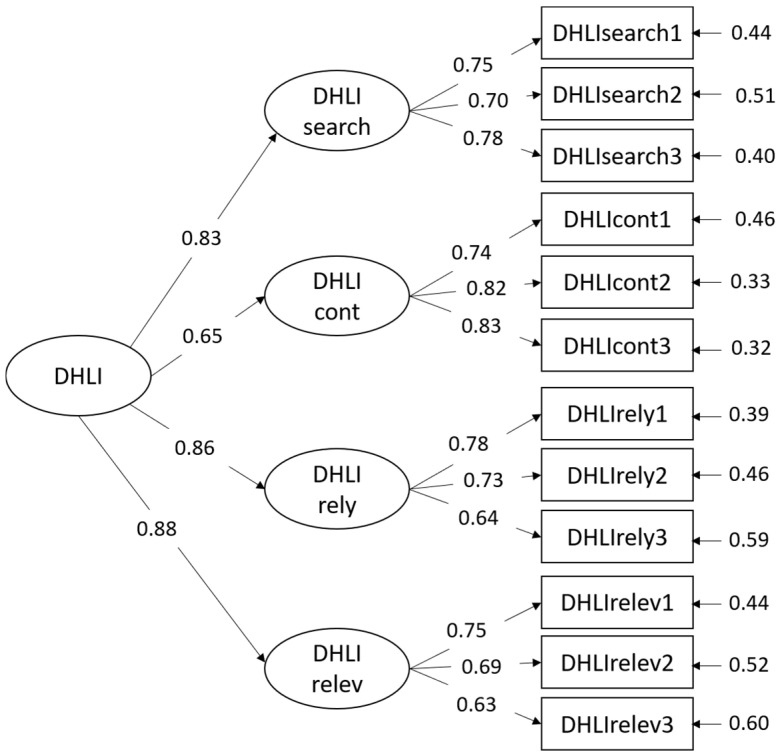
Structural Equation Models (SEMs).

**Figure 2 ijerph-19-06247-f002:**
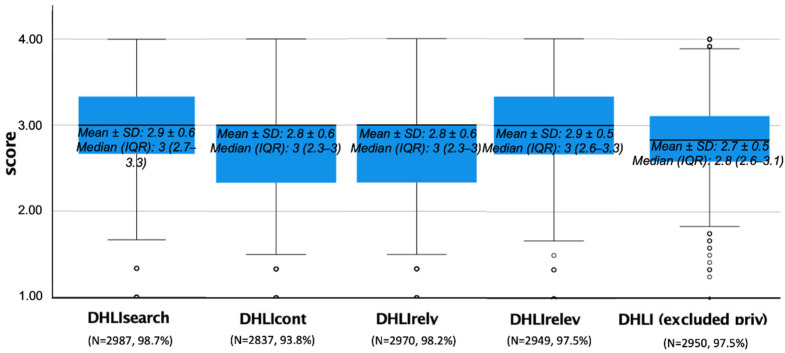
Descriptive analysis of the subscales and scale scores: box plots.

**Table 1 ijerph-19-06247-t001:** COVID-19 Digital Health Literacy Instrument (DHLI): item responses (*n* = 3025).

Area (Subscales)	Items	Missing *n* (%)	Very Difficult *n* (%)	Difficult *n* (%)	Easy *n* (%)	Very Easy *n* (%)	Mean ± SD	Median (IQR)
DHLI—information searching (DHLIsearch)	DHLIsearch1	35 (1.2)	81 (2.7)	806 (26.6)	1671 (55.2)	432 (14.3)	2.8 ± 0.7	3 (2–3)
	DHLIsearch2	43 (1.4)	25 (0.8)	359 (11.9)	1874 (62.0)	724 (23.9)	3.1 ± 0.6	3 (3–3)
	DHLIsearch3	40 (1.3)	133 (4.4)	938 (31.0)	1470 (48.6)	444 (14.7)	2.7 ± 0.7	3 (2–3)
DHLI—adding self-generated content (DHLIcont)	DHLIcont1	197 (6.5)	62 (2.0)	600 (19.8)	1760 (58.2)	406 (13.4)	2.9 ± 0.6	3 (3–3)
	DHLIcont2	188 (6.3)	136 (4.5)	762 (25.2)	1481 (49.0)	458 (15.1)	2.8 ± 0.8	3 (2–3)
	DHLIcont3	192 (6.3)	138 (4.6)	856 (28.3)	1472 (48.7)	367 (12.1)	2.7 ± 0.7	3 (2–3)
DHLI—evaluating reliability (DHLIrel)	DHLIrely1	56 (1.9)	220 (7.3)	1145 (37.9)	1280 (42.3)	324 (10.7)	2.6 ± 0.8	3 (2–3)
	DHLIrely2	61 (2.0)	158 (5.2)	945 (31.2)	1375 (45.5)	486 (16.1)	2.7 ± 0.8	3 (2–3)
	DHLIrely3	61 (2.0)	43 (1.4)	389 (12.9)	1725 (57.0)	807 (26.7)	3.1 ± 0.6	3 (3–4)
DHLI—determining relevance (DHLIrelev)	DHLIrelev1	78 (2.6)	19 (0.6)	438 (14.5)	1994 (65.9)	496 (16.4)	3.0 ± 0.6	3 (3–3)
	DHLIrelev2	87 (2.9)	41 (1.4)	681 (22.5)	1801 (59.5)	415 (13.7)	2.9 ± 0.6	3 (3–3)
	DHLIrelev3	77 (2.5)	73 (2.4)	549 (18.1)	1755 (58.0)	571 (18.9)	2.9 ± 0.7	3 (3–3)
**Area (Subscales)**	**Items**	**Missing** ***n* (%)**	**Often** ***n* (%)**	**Several Times** ***n* (%)**	**Once** ***n* (%)**	**Never** ***n* (%)**	**Mean ± SD**	**Median (IQR)**
DHLI—protecting privacy (DHLIpriv)	DHLIpriv1	271 (9.0)	181 (6.0)	702 (23.2)	651 (21.5)	1220 (40.3)	3.1 ± 1.0	3 (2–4)
	DHLIpriv2	254 (8.4)	112 (3.7)	371 (12.3)	483 (16)	1805 (59.7)	3.4 ± 0.9	4 (3–4)
	DHLIpriv3	257 (8.5)	17 (0.6)	93 (3.1)	234 (7.7)	2424 (80.1)	3.8 ± 0.5	4 (4–4)

SD: standard deviation; IQR: interquartile range.

**Table 2 ijerph-19-06247-t002:** COVID-19 Digital Health Literacy Instrument (DHLI): Spearman correlation analysis (*n* = 3025).

ITEMS	DHLI Search1	DHLI Search2	DHLI Search3	DHLI Cont1	DHLI Cont2	DHLI Cont3	DHLI Rely1	DHLI Rely2	DHLI Rely3	DHLI Relev1	DHLI Relev2	DHLI Relev3	DHLI Priv1	DHLI Priv2
DHLIsearch2	0.51 °													
DHLIsearch3	0.56 °	0.56 °												
DHLIcont1	0.33 °	0.39 °	0.36 °											
DHLIcont2	0.28 °	0.31 °	0.32 °	0. 58 °										
DHLIcont3	0.32 °	0.34 °	0.36 °	0.57 °	0.68 °									
DHLIrely1	0.55 °	0.36 °	0.45 °	0.33 °	0. 29 °	0.32 °								
DHLIrely2	0.37 °	0.30 °	0.35 °	0.29 °	0.26 °	0.27 °	0.58 °							
DHLIrely3	0.34 °	0.38 °	0.34 °	0.31 °	0.28 °	0.29 °	0.44 °	0.48 °						
DHLIrelev1	0.44 °	0.43 °	0.44 °	0.39 °	0.34 °	0.33 °	0.48 °	0.45 °	0.45 °					
DHLIrelev2	0.33 °	0.33 °	0.36 °	0.35 °	0.33 °	0.33 °	0.34 °	0.31 °	0.37 °	0.49 °				
DHLIrelev3	0.30 °	0.28 °	0.32 °	0.33 °	0.30 °	0.29 °	0.34 °	0.33 °	0.36 °	0.42 °	0.55 °			
DHLIpriv1	0.14 °	0.15 °	0.16 °	0.18 °	0.17 °	0.22 °	0.16 °	0.14 °	0.12 °	0.18 °	0.16 °	0.15 °		
DHLIpriv2	0.06 °	0.04 °	0.03 ^#^	0.03 ^#^	−0.003 ^#^	0.04 ^#^	0.06 °	0.07 °	0.07 °	0.03 ^#^	0.08 °	0.04 °	0.14 °	
DHLIpriv3	0.07 °	0.11 °	0.04 *	0.042 *	0.01 ^#^	0.01 ^#^	0.04 *	0.09 °	0.11 °	0.08 °	0.10 °	0.06 °	0.14 °	0.42 °

° *p* < 0.001; * 0.001 < *p* < 0.05; ^#^
*p* ≥ 0.05.

**Table 3 ijerph-19-06247-t003:** Internal consistency of the items: Cronbach’s alpha.

Items	For the Entire Scale	For the Entire Scale If Item Deleted	For the Entire Scale Excluding DHLIpriv	By Subscales
DHLIsearch1	0.847	0.835	0.881	0.783
DHLIsearch2		0.837		
DHLIsearch3		0.834		
DHLIcont1		0.836		0.834
DHLIcont2		0.838		
DHLIcont3		0.836		
DHLIrely1		0.832		0.758
DHLIrely2		0.836		
DHLIrely3		0.837		
DHLIrelev1		0.834		0.739
DHLIrelev2		0.837		
DHLIrelev3		0.840		
DHLIpriv1		0.857	-	0.392
DHLIpriv2		0.864		
DHLIpriv3		0.853		

**Table 4 ijerph-19-06247-t004:** Principal component analysis (varimax rotation).

ITEMS	MODEL 1 Components—by the Data (Based on Eighenvalues) *				MODEL 2 Components—by the Data Excluding DHLIpriv (Based on Eighenvalues) ^§^			MODEL 3 Components—by Literature (5 Components) °					MODEL 4 Components—by Literature Excluding DHLIpriv (4 Components) ^#^			
	1	2	3	4	1	2	3	1	2	3	4	5	1	2	3	4
DHLIsearch1	**0.796**	0.166	0.147	0.034	**0.792**	0.181	0.152	**0.797**	0.166	0.149	0.03	0.052	0.127	**0.739**	0.352	0.103
DHLIsearch2	**0.717**	0.098	0.267	0.07	**0.716**	0.1	0.274	**0.731**	0.115	0.29	0.104	−0.091	0.204	**0.788**	0.114	0.164
DHLIsearch3	**0.747**	0.159	0.25	0.01	**0.744**	0.165	0.255	**0.754**	0.166	0.261	0.022	−0.009	0.198	**0.778**	0.202	0.184
DHLIcont1	0.261	0.216	0.745	0.014	0.244	0.22	**0.758**	0.261	0.219	**0.748**	0.013	0.059	**0.755**	0.233	0.152	0.195
DHLIcont2	0.163	0.202	0.831	−0.029	0.146	0.196	**0.845**	0.158	0.2	**0.826**	−0.042	0.105	**0.860**	0.113	0.15	0.149
DHLIcont3	0.216	0.164	0.83	0.025	0.207	0.162	**0.836**	0.208	0.158	**0.821**	0.004	0.142	**0.846**	0.181	0.145	0.121
DHLIrely1	**0.638**	0.433	0.103	0.025	**0.635**	0.443	0.09	**0.611**	0.403	0.066	−0.052	0.341	0.157	**0.363**	**0.734**	0.116
DHLIrely2	0.493	**0.528**	0.037	0.073	0.49	**0.54**	0.03	0.461	**0.492**	−0.01	−0.021	0.41	0.135	0.136	**0.851**	0.124
DHLIrely3	0.41	**0.56**	0.088	0.088	**0.391**	**0.575**	0.1	0.397	**0.546**	0.07	0.048	0.201	0.156	0.159	**0.652**	0.295
DHLIrelev1	0.431	**0.572**	0.236	0.042	0.427	**0.577**	0.23	0.425	**0.565**	0.227	0.019	0.136	0.222	**0.341**	0.441	**0.459**
DHLIrelev2	0.12	**0.74**	0.26	0.088	0.11	**0.743**	0.265	0.134	**0.756**	0.279	0.118	−0.07	0.192	0.193	0.156	**0.816**
DHLIrelev3	0.082	**0.779**	0.196	0.005	0.065	**0.773**	0.209	0.091	**0.788**	0.207	0.022	−0.031	0.149	0.116	0.202	**0.810**
DHLIpriv1	0.078	0.089	**0.329**	**0.343**	-	-	-	0.007	0.011	0.226	0.148	**0.836**	-	-	-	-
DHLIpriv2	0.025	0.006	−0.018	**0.821**	-	-	-	0.032	0.013	−0.012	**0.828**	0.075	-	-	-	-
DHLIpriv3	0.034	0.079	−0.018	**0.821**	-	-	-	0.044	0.09	−0.007	**0.835**	0.047	-	-	-	-
Explained variance	35.9%	9.9%	8.7%	6.8%	43.8%	11.2%	8.6%	35.9%	9.9%	8.7%	6.8%	5.9%	43.8%	11.2%	8.6%	7.2%

* Explained variance: 61.4%, Kaiser–Meyer–Olkin test: 0.884 (excellent), Barlett sphericity test: *p* < 0.001; ^§^ Explained variance: 63.6%, Kaiser–Meyer–Olkin test: 0.892 (excellent), Barlett sphericity test: *p* < 0.001; ° Explained variance: 67.3%, Kaiser–Meyer–Olkin test: 0.884 (excellent), Barlett sphericity test: *p* < 0.001; ^#^ Explained variance:70.4%, Kaiser–Meyer–Olkin test: 0.892 (excellent), Barlett sphericity test: *p* < 0.001.

**Table 5 ijerph-19-06247-t005:** Fit statistics of the confirmatory factor analysis (*n* = 2770).

Fit Statistics		MODEL A (1 Factor)	MODEL B (4 Factors)	MODEL C (4 + 1 Factors)
Chi2		3826.24	741.18	784.64
GDL		54	48	50
Overall Model Fit	RMSEA (90% CI)	0.159 (0.155–0.163)	0.072 (0.068–0.077)	0.073 (0.068–0.077)
	SRMR	0.08310	0.03981	0.04205
Model comparison	GFI	0.8128	0.957	0.955
	CFI	0.8865	0.977	0.976
	NNFI	0.8613	0.968	0.968
Model parsimony	PNFI	0.7240	0.7093	0.7379
	PGFI	0.5627	0.5891	0.6121

## Data Availability

The data presented in this study is available on request from the corresponding author.

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
