# Peer review of "Validation of the COVID-19 Digital Health Literacy Instrument in the Italian Language: A Cross-Sectional Study of Italian University Students"

_ijerph, 2022, doi:10.3390/ijerph19106247_

Round 1

Reviewer 1 Report

The authors present in the article as their main objective is to validate the Italian version of the Digital Health Literacy Instrument developed by van der Vaart and Drossaert and adapted to the COVID-19 pandemic by other authors.
The article is well taken care of and properly supported by reference documentation.
However, there are some considerations to take into account:
1) there seems to be some confusion in the definition of the acronyms DHLI (defined in the abstract) and DHL, as they both refer to the same description.
2) A table with the acronyms should be included since their diversity makes it difficult to understand and frame them.
3) The introduction is too long, suggesting creating two subsections: framework and related work.

Author Response

We thank you for your thoughtful suggestions and insights, which have helped us improved the paper significantly. We are also grateful for your positive feed-backs.

The manuscript has been rechecked and the necessary changes have been made in accordance with your suggestions. The responses to all comments have been prepared and given below.

  1. There seems to be some confusion in the definition of the acronyms DHLI (defined in the abstract) and DHL, as they both refer to the same description

Response: Thanks for the very useful suggestion, which has helped us correct this confusing element. The acronym DHL refers to the concept of digital health literacy, while DHLI refers to the scale “Digital Health Literacy Instrument”. We better defined the DHLI acronym in the abstract and we checked the use of both acronyms throughout the manuscript.

  1. A table with the acronyms should be included since their diversity makes it difficult to understand and frame them

Response: Thanks for the very useful suggestion, which has helped us improved the paper. We added a table with the acronyms’ definition in the Supplementary materials.

  1. The introduction is too long, suggesting creating two subsections: framework and related work

Response: Thanks for the very useful suggestion, which has helped us improved the paper significantly. We divided the introduction into two subsections: 1) “Health literacy and COVID-19” which presents the framework of the paper and 2) “Digital Health Literacy assessment” which describes the literature about the topic of the paper

Reviewer 2 Report

I would like to congratulate the authors on an insightful and useful study. The study on the evaluation of the COVID-19 Digital Health Literacy instrument will certainly be of tangible benefit. 
The introduction was prepared in a clear manner. The authors have well described the state of research on the Digital Health Literacy (DHLI) scale.

The text was methodologically well prepared.
The study was reliably conducted using statistical methods. 
The discussion was conducted in a critical manner. 

The results of the study should be published so that researchers in other countries can benefit from the experience of the Italian researchers. 

Author Response

We thank you for the positive feed-backs

Reviewer 3 Report

This review is interesting and important for understanding Validation of the COVID-19 Digital Health Literacy Instrument in the Italian language. But several points must be improved.

Title

(Comment 1) I recommend authors to change 'results from a university studients' survey' into 'a cross-sectional study of Italian university studients'

Method

(Comment 2) I recommend authors to re-write "Materials and Methods Section". Survey form must be presented as a supplementary file. This could be helpful for future researcher.

e.g.

Materials and Methods

2.1. Method for survey

2.1.1. Survey sampling (e.g. sampling process)

2.1.2. Development of the survey form (e.g. translate to italian)

2.1.3. Survey distribution

2.1.4. Data collection

2.1.5. Statistical analysis analysis

(Comment 3) I recommend authors to supplement study limitation in "Discussion section".

Author Response

We thank you for your thoughtful suggestions and insights, which have helped us improved the paper significantly. We are also grateful for your positive feed-backs.

The manuscript has been rechecked and the necessary changes have been made in accordance with your suggestions. The responses to all comments have been prepared and given below.

  1. I recommend authors to change 'results from a university studients' survey' into 'a cross-sectional study of Italian university studients'

Response: Thanks for the very useful suggestion. We changed the title as suggested.

  1. I recommend authors to re-write "Materials and Methods Section". Survey form must be presented as a supplementary file. This could be helpful for future researcher.

e.g. Materials and Methods: 2.1. Method for survey, 2.1.1. Survey sampling (e.g. sampling process), 2.1.2. Development of the survey form (e.g. translate to italian), 2.1.3. Survey distribution, 2.1.4. Data collection, 2.1.5. Statistical analysis analysis

Response: Thanks for the very useful suggestion, which has helped us improved the paper significantly. We re-write "Materials and Methods Section" as suggested.

  1. I recommend authors to supplement study limitation in "Discussion section".

Response: Thanks for the very useful suggestion. We discussed study limitations better.

Round 2

Reviewer 3 Report

After reviewing the author responses, the author successfully addressed most of the comments and suggestions. Regarding method section, sampling and survey distribution process is too short. (line 169-176) I recommend authors to supplement this point in the method section.

Author Response

Thanks for the very useful suggestion, which has helped us improved the paper significantly. As suggested, we supplemented the Method section by giving more information about the sampling and survey distribution process.